# Resonance Control of VO$_2$ Thin-Film-Based THz Double-Split Rectangular Metamaterial According to Aspect Ratio

**Eui Su Lee** [1] and **Han-Cheol Ryu** [2,*]

1 Terahertz Research Section, Electronics and Telecommunications Research Institute, Daejeon 34129, Republic of Korea
2 Division of AI Informatics, Sahmyook University, Seoul 01795, Republic of Korea
* Correspondence: hcryu@syu.ac.kr

**Abstract:** The resonance characteristics of a double-split rectangular metamaterial based on a vanadium dioxide (VO$_2$) thin film were controlled according to the aspect ratio of the rectangle in the terahertz (THz) frequency region. The VO$_2$ thin film line was etched between the double-split rectangular gaps so that the resonance band could be switched by varying the characteristics of the VO$_2$ thin film. When the VO$_2$ thin film is in an insulator state, the rectangle is separated and resonates individually; thus, it resonates in the high-frequency band. When the VO$_2$ thin film changes from an insulator to a conductor with a change in the temperature, the divided rectangles are electrically connected to operate as a single resonator, and the resonant frequency shifts to a low-frequency band. Varying the aspect ratio of the rectangle changes the resonant frequency and resonance strength of the double-split rectangular metamaterial. If the aspect ratio is increased by fixing the width of the unit cell of the metamaterial and adjusting the height, the resonant frequency is lowered in all situations, regardless of the state of the VO$_2$ thin film and the polarization of the incident THz wave. The resonant frequency and resonance strength of the double-split rectangular metamaterial proposed in this paper could be controlled stably through a change in only the aspect ratio, not the overall unit cell size. The proposed double-split rectangular metamaterial based on an etched VO$_2$ thin film is expected to be essential for THz tag, sensing, and wireless communication applications.

**Keywords:** metamaterials; terahertz (THz); vanadium dioxide; double-split rectangular; aspect ratio; THz-TDS; transmittance; band switching





## 1. Introduction

Terahertz (THz) technology is actively used in academia and research in spectroscopy and imaging systems. Several studies have been conducted on various sensing fields and wireless communication applications [1–6]. Owing to the explosive increase in data and the need to transmit them using wireless communication, wireless communication using the THz frequency, which is most suitable for large-capacity wireless communication, has attracted considerable attention. To utilize THz wireless communication technology, it is crucial not only to develop various THz devices but also to develop THz devices that can control electromagnetic wave response characteristics [7–10]. In addition, a device with tunable resonance characteristics needs to be developed to apply THz technology actively to various application fields, such as THz tags and sensors. However, the THz devices, which are currently not sufficiently developed, are delaying the active use of the THz tag, sensor, and THz wireless communication technology.

In the THz frequency band, natural materials do not have the high dielectric constant and low loss required for the development of THz devices. Therefore, development of THz devices using metamaterials whose electromagnetic resonance can be controlled has received considerable attention [11–16]. Metamaterials consist of periodic unit cells of metal structures that are smaller than the wavelength of the operating electromagnetic

wave. As the resonance of the metamaterial is determined by the LC resonance owing to the structure of the unit cell, the desired electromagnetic properties of the THz device can be obtained by changing the periodic structure of the metamaterial [17,18]. Metamaterials can be used to develop new types of THz devices and application systems. In addition, the controllable resonance of artificially engineered metamaterials may enable realization of novel THz devices for various THz applications.

Several studies have been conducted on tunable metamaterials with various functions in the THz band. In particular, several studies have been reported on tunability of THz metamaterials' properties using semiconductors, two-dimensional materials, such as graphene, and various functional materials [19–21]. Among the tunable functional materials, vanadium dioxide ($VO_2$) has attracted considerable attention for applications in tunable THz wave devices owing to its easy fabrication and high tunability [22–28]. As the dielectric properties of a $VO_2$ thin film can be controlled by its phase transition, the resonance properties of metamaterials composed of a $VO_2$ substrate can be controlled. $VO_2$ thin films are suitable for developing various THz tunable devices, as they provide excellent modulation and switching characteristics in the broad THz region because free carriers are rapidly generated during the phase shift process. However, a tunable metamaterial using an entire $VO_2$ thin film as a substrate has a large transmission loss, and implementing evident band-changing characteristics is challenging. In addition, the quality factors of metamaterials need to be controlled to improve the performance of THz devices based on metamaterials and apply them to various systems. The band-switching and tunable properties of etched $VO_2$ thin-film-based square-structured THz metamaterials have been reported in previous studies [22]. In this study, the resonant frequency was controlled by changing the overall size of the unit cell square. However, the change in unit cell size directly affects the overall size of the device, and the quality factor of metamaterials becomes difficult to control. An asymmetric structure can be created by breaking the symmetry of the square structure to control the quality factor of the metamaterial. However, owing to this asymmetry, unwanted new resonance occurs in the metamaterial, thereby changing the existing resonance properties [24]. This paper proposes a double-split rectangular metamaterial based on an etched $VO_2$ thin film with tunable THz electromagnetic response. The etched $VO_2$ thin film line was positioned between the double-split gaps to secure the band-switching characteristics of the metamaterial. Furthermore, the resonant frequency and quality factor of the metamaterial were successfully controlled by changing the aspect ratio of the rectangle.

## 2. Design and Simulation

### 2.1. Design of Double-Split Rectangular Metamaterial

Figure 1 shows the schematics of a double-split rectangular metamaterial with various aspect ratios based on an etched $VO_2$ thin film. We fixed the width of the rectangle and changed its height to control the resonance characteristics of the metamaterial. The $VO_2$ thin film was etched to a width of 10 μm, which is twice that of the 5 μm gap; the $VO_2$ thin film line was positioned entirely in the double-split gaps. The etched thin film changes from an insulator to a conductor with a change in the temperature, thereby switching the resonance band of the metamaterial. Suppose that the etched $VO_2$ thin film line between the two gaps is in the insulator phase; the rectangles are separated by the gaps so that the two resonators act separately and resonate in the high-frequency band. When the $VO_2$ thin-film line changes to the conductor phase with a change in the temperature, the rectangle separated by the gap is electrically connected to form a resonator, and resonance occurs in the low-frequency band. The primary cell structure of the metamaterial was rectangular, the width was fixed at 60 μm, and the height was varied to control the resonant frequency and quality factor. The width of the metal resonator inside the unit cell was 50 μm, and the height was changed such that the distance between the metal resonators inside each cell was maintained at 5 μm. In addition, the gaps dividing the rectangular metal resonator into two and the width of the metal line were set to 5 μm for easy implementation using

general photolithography technology. The VO$_2$ thin film line was etched to 10 μm to ensure a stable connection between the metal resonators. As the proposed metamaterial has different shapes in the horizontal and vertical directions, the electromagnetic response characteristics vary depending on the polarization state of the incident THz electric field. In Figure 1, mode 1 indicates horizontal polarization, where the polarity of the electric field is parallel to the etched VO$_2$ thin film line; mode 2 indicates vertical polarization, where the polarity is perpendicular to the line. The resonant frequency and quality factor can be easily tuned by controlling the aspect ratio through a change in the height of the rectangular metal structures constituting the metamaterial.

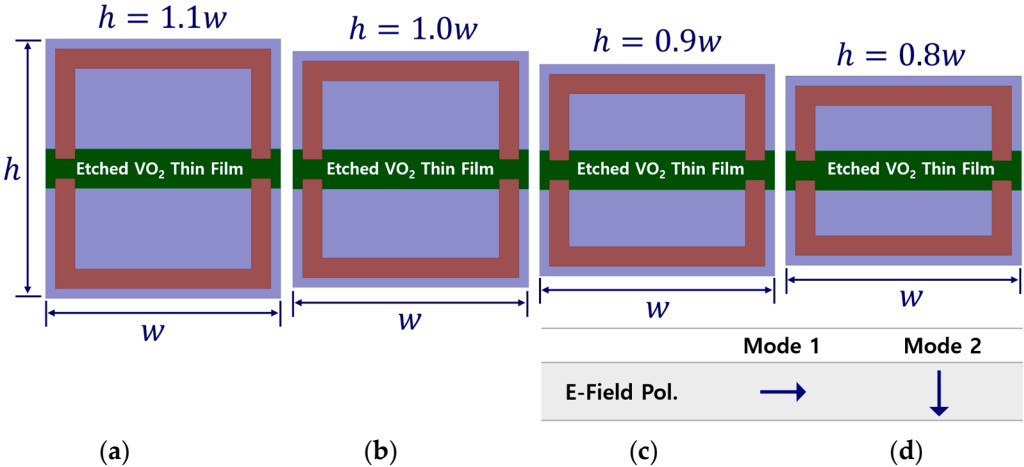

**Figure 1.** Schematics of a double-split rectangular metamaterial with various aspect ratios based on an etched VO$_2$ thin film. (**a**) Aspect ratio = 1.1 (h = 1.1 w), (**b**) aspect ratio = 1.0 (h = w), (**c**) aspect ratio = 0.9 (h = 0.9 w), (**d**) aspect ratio = 0.8 (h = 0.8 w).

### 2.2. Simulation of a Double-Split Rectangular Metamaterial

The proposed double-split rectangular metamaterial was simulated using a 3D electromagnetic field simulator, HFSS (high-frequency structure simulator), to calculate its electromagnetic response in the THz band. The THz transmission characteristics of the proposed metamaterial depend on the change in the conductivity of the VO$_2$ thin film, and the conductivity of the VO$_2$ thin film depends on the temperature. The conductivity of the VO$_2$ thin film was set to 20 S/m in the insulator state and 1,000,000 S/m in the conductor state. The conductivity was confirmed by measurement using a four-probe measurement [25]. The dielectric constants of the VO$_2$ thin film and Al$_2$O$_3$ were fixed at 9.1 and 9.4, respectively. The frequency dependence of the material is not considered. The width of the double-split rectangular metamaterial was 60 μm, and the height was changed from 51 μm to 75 μm in intervals of 3 μm for the simulation. The aspect ratio of the rectangular metamaterial was increased from 0.85 to 1.25 in increments of 0.05. The spacing and width of the metal structures, and the spacing of the metal structures between the unit cells, were fixed at 5 μm for easy fabrication using conventional photolithography. As the proposed metamaterial has different structures in the horizontal and vertical directions, the polarization state of the incident THz electric field was simulated by dividing it into horizontal and vertical polarizations. The port was set so that only one mode was incident with a horizontal (parallel to the etched VO$_2$ line) and vertical (perpendicular to the etched VO$_2$ line) polarization electric field for mode 1 and mode 2, respectively.

Figure 2 shows the transmittance of the THz electromagnetic wave through the double-split rectangular metamaterial in mode 1, which is a state of electric field polarization parallel to the VO$_2$ thin film line, according to the aspect ratio. As the direction of the incident THz electric field in mode1 is the same as the direction of the etched VO$_2$ thin film line pattern, resonance mainly occurs because of the double-split rectangular horizontal metal structure parallel to the etched VO$_2$ thin film line. When VO$_2$ is in the insulator state,

the $VO_2$ thin film line does not affect resonance; therefore, only resonance owing to the metal structure occurs, as shown in Figure 2a. However, when $VO_2$ changes to the conductor state, the $VO_2$ thin film line directly affects resonance, resulting in strong resonance at low frequencies, as shown in Figure 2b. The intense low-frequency resonance generated from the $VO_2$ line and the high- frequency resonance generated from the metal structure combine to show a bandpass characteristic in the middle band of the two resonances. As the aspect ratio of the double-split rectangles constituting the metamaterial increases, the structure becomes larger as the width of the rectangle is fixed, resulting in lower resonant frequencies in both the insulator and conductor phases of the $VO_2$ thin film. Increasing the aspect ratio of the rectangle in the $VO_2$ insulator state tends to increase the quality factor of the metamaterial, as the resonance bandwidth decreases by a more significant proportion than the decrease in the resonant frequency. On the other hand, when the aspect ratio of the rectangle increases in the $VO_2$ conductor state, the resonant frequency decreases. However, the resonance bandwidth hardly changes; therefore, the quality factor of the metamaterial tends to decrease. When the $VO_2$ thin film is in insulating state, the fundamental resonance of the double-split rectangular metal structure occurs in the frequency band of 0.57 THz to 0.84 THz. When the $VO_2$ thin film changes to the conductive state, the fundamental resonance of the metamaterial shifts to the frequency band of 1.25 THz to 1.6 THz. As the aspect ratio increases from 0.85 to 1.25, the transmittance of the metamaterial in the passband increases from 47.3% to 62.3%.

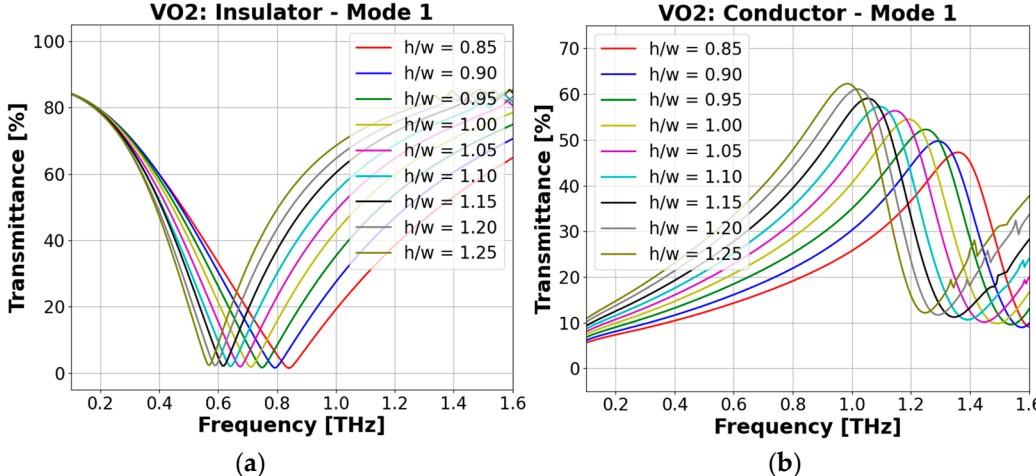

**Figure 2.** Transmittance of the double-split rectangular metamaterial in mode 1 with the electric field polarization parallel to the $VO_2$ thin film line according to the aspect ratio. (**a**) $VO_2$ in the insulator state and (**b**) $VO_2$ in the conductor state.

As shown in Figure 2, as the $VO_2$ thin film constituting the metamaterial changes from an insulator to a conductor, its resonant frequency shifts from the 0.55 to 0.85 THz band to the 1.25 to 1.6 THz band. Moreover, a new resonance occurred in the low-frequency band, changing the dip in the 0.55 to 0.85 THz band to the peak in the 1.0 to 1.4 THz band. Figure 3 shows the surface current density generated in the resonance state of each phase to analyze the change in the resonance properties of the double-split rectangular metamaterial with the change in the $VO_2$ thin film from the insulator to conductor phase. The aspect ratio of the rectangle is 1.25. Figure 4a,b show the surface current densities induced in the metamaterial at the resonance frequencies of 0.57 THz when the $VO_2$ film is in the insulator state and 1.25 THz when the $VO_2$ film is in the conductor state, respectively. As shown in Figure 3a, owing to the insulating properties of the $VO_2$ thin film, the resonances generated in the two metal resonators are separated and have a relatively long resonance path. Figure 3b shows that the two spatially separated metal resonators are electrically connected by a $VO_2$ thin film on the conductor. The surface currents generated by the two fundamental resonators are directed toward the $VO_2$ thin film. As the two surface

current paths are divided, the resonant path length of the connected resonator is reduced to approximately half of that of the separated resonator, and the resonant frequency is approximately doubled.

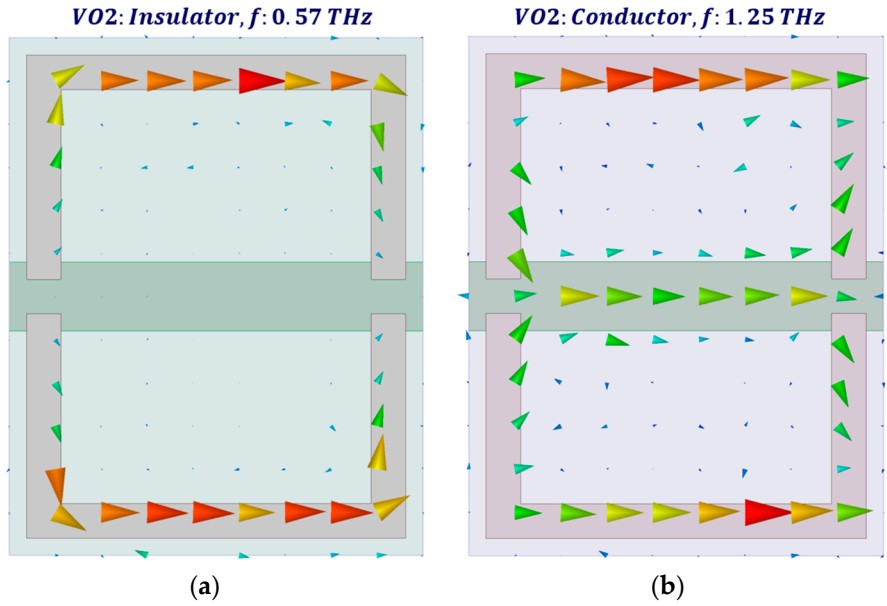

**Figure 3.** Surface current density of the double-split rectangular metamaterial with an aspect ratio of 1.25 operating in mode 1, (**a**) insulator phase (f = 0.57 THz), (**b**) conductor phase (f = 1.25 THz).

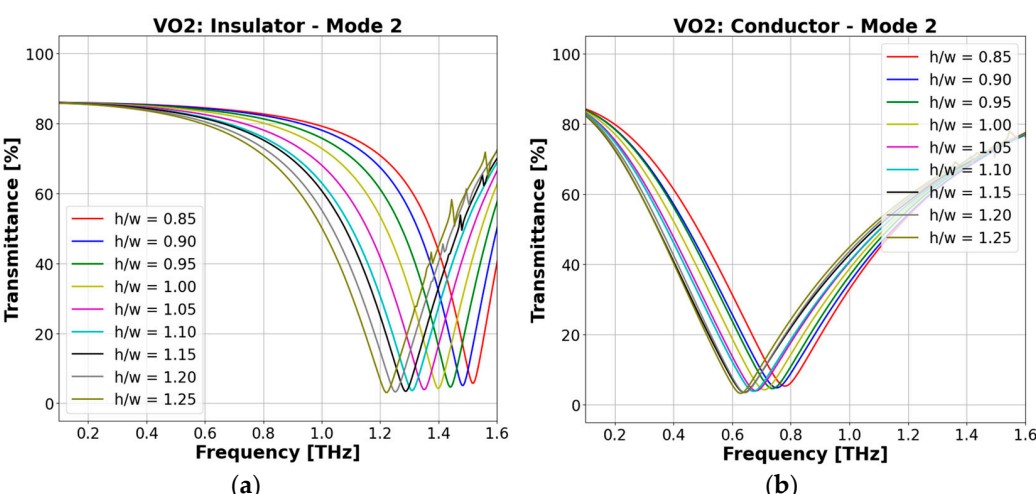

**Figure 4.** Transmittance of the double-split rectangular metamaterial in mode 2 with electric field polarization perpendicular to the $VO_2$ thin film line according to the aspect ratio. (**a**) $VO_2$ in the insulator state and (**b**) $VO_2$ in the conductor state.

Figure 4 shows the transmittance of the THz electromagnetic wave through the double-split rectangular metamaterial in mode 2, which is a state of electric field polarization perpendicular to the $VO_2$ thin film line, according to the aspect ratio. As the direction of the THz electric field incident in mode 2 is perpendicular to the direction of the etched $VO_2$ thin film line pattern, resonance mainly occurs because of the double-split rectangular vertical metal structure perpendicular to the etched $VO_2$ thin film. When the $VO_2$ thin film is in the insulator state, it does not affect resonance; therefore, only resonance owing to the metal structure occurs. Owing to the insulating properties of the $VO_2$ thin film, the resonances generated in the two metal resonators are separated, and resonance occurs in the high-frequency band of 1.2 THz or higher. However, when the $VO_2$ thin film changes to

the conductive state, it directly affects resonance. Two spatially separated metal resonators are electrically connected to generate a resonance in the low-frequency band of 0.8 THz or lower. Therefore, when the VO$_2$ thin film is in the insulator phase, the resonant frequency is approximately twice that in the conductor phase. If the aspect ratio of the rectangle increases in the VO$_2$ insulator state, the resonant frequency decreases, and the bandwidth increases; thus, the quality factor decreases. However, when the aspect ratio of the rectangle increases in the VO$_2$ conductor state, the resonant frequency decreases slightly. However, the resonant bandwidth does not change significantly, and thus, the quality factor decreases.

Figure 5 shows the change in the resonant frequency and quality factor of the double-split rectangular metamaterial in modes 1 and 2 with VO$_2$ thin film in the insulator and conductor states according to the aspect ratio of the rectangle. As shown in Figure 5a, if the width of the rectangle is fixed and the aspect ratio increases, the size of the basic structure constituting the unit cell increases; thus, the resonant frequency increases in all cases. In the insulator state, the VO$_2$ thin film resonates in a high-frequency band when the incident THz wave is vertically polarized and resonates in a low-frequency band when the incident THz wave is horizontally polarized. The length of the metal structure, which is in the same direction as the polarization direction of the incident THz wave, primarily determines the resonant frequency. On the other hand, when the VO$_2$ thin film changes to the conductor state, the film directly affects resonance. When the THz wave is vertically polarized, VO$_2$ connects the two resonators to lengthen the metal structure in the same direction as the polarization direction so that the metamaterial resonates in a low-frequency band. As the VO$_2$ thin film is in the same direction as the polarization when the THz wave is horizontally polarized, a new direct resonance path is formed, and the two resonance paths become three. The new resonant path is reduced in length, and resonance occurs in the high-frequency band. When the VO$_2$ thin film is in the insulator state, and the THz wave is vertically polarized, the quality factor is the highest. When the aspect ratio of the metamaterial increases from 0.85 to 1.25, the quality factor decreases from 6.6 to 3.5, as shown in Figure 5b. When the VO$_2$ thin film is in the conductive state, the quality factor decreases as the aspect ratio of the metamaterial increases, regardless of the polarization of the THz wave. When the aspect ratio increases from 0.85 to 1.25, the quality factor decreases from 2.4 to 1.7 in mode 1 and decreases from 1.6 to 1.4 in mode 2. Only when the VO$_2$ film is insulative and the polarization of the THz wave is parallel to the VO$_2$ thin film line does the quality factor increase proportionally to the aspect ratio of the metamaterial. This is because, as the aspect ratio of the metamaterial increases, the resonant frequency of the metamaterial decreases, and the resonance intensity increases.

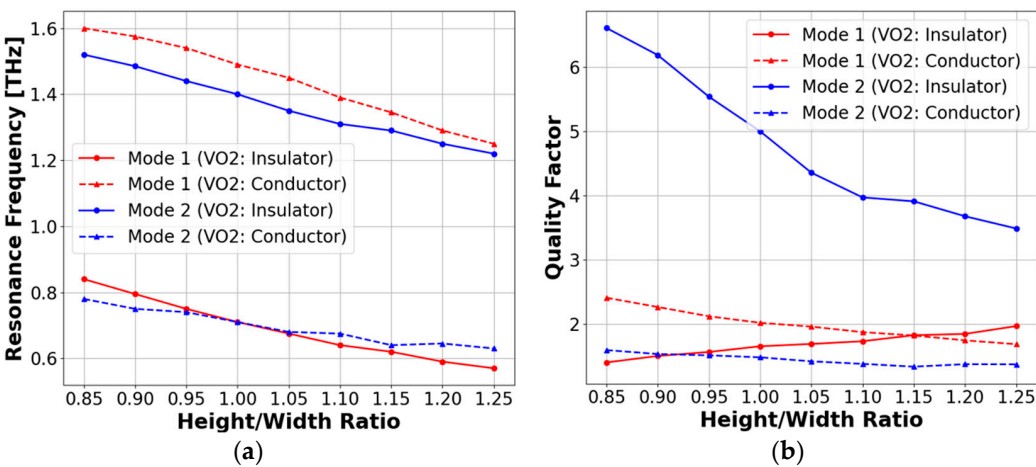

**Figure 5.** (**a**) Resonant frequency and (**b**) quality factor of the double-split rectangular metamaterial in modes 1 and 2 with the VO$_2$ thin film in the insulator and conductor states according to the aspect ratio of the rectangle.

## 3. Fabrication and Measurement

### 3.1. Fabrication of the Double-Split Rectangular Metamaterial

Figure 6 shows photographs of the double-split rectangular metamaterials with various aspect ratios fabricated based on etched VO$_2$ thin films. Figure 6a,b show the photographs of a metamaterial with a rectangle aspect ratio of 1.0 at 20× and 80× magnification, respectively, and Figure 6c,d are photographs with an aspect ratio of 0.9 at 20× and 80× magnification, respectively. The tunable range of metamaterials needs to be increased by depositing VO$_2$ thin films to maximize the rate of change of electrical conductivity. Therefore, a single-phase VO$_2$ thin film was grown on a single-crystal Al$_2$O$_3$ (0001) substrate using an ion-reactive radio-frequency (RF) sputtering method. The deposition time was adjusted to deposit the VO$_2$ thin film of thickness 100 nm. The thickness of the thin films was confirmed using cross-sectional scanning electron microscope. The deposited VO$_2$ thin film was etched in the form of a line suitable for placement between the gaps of the double-split rectangle. A gold electrode (200 nm) with a Ti adhesive layer (10 nm) was deposited on top of the etched VO$_2$ thin film via DC sputtering method to form a metallic double-split rectangular structure. The designed VO$_2$ thin film lines and metal double-split rectangular structures were patterned using conventional photolithography and lift-off processes.

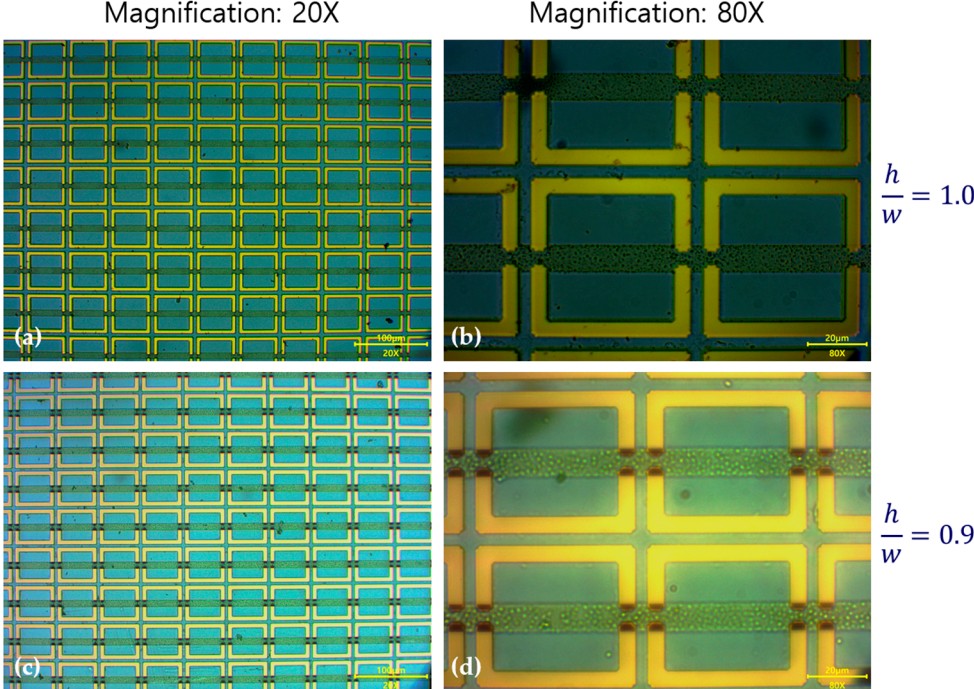

**Figure 6.** Photographs of the double-split rectangular metamaterial with various aspect ratios based on etched VO$_2$ thin films. (**a**) Magnification: 20×, h/w = 1.0, (**b**) magnification: 80×, h/w = 1.0, (**c**) magnification: 20×, h/w = 0.9, (**d**) magnification: 80×, h/w = 0.9.

### 3.2. Measurement of a Double-Split Rectangular Metamaterial

Two metamaterials with the aspect ratios of 1.0 and 0.9 were fabricated using conventional photolithography to measure the change in the THz wave transmittance and resonance characteristics of the double-split rectangular metamaterial according to the aspect ratio. The fabricated metamaterial was measured using a time-domain spectroscopy (THz-TDS) system, a TAS7400 model from Advantest Corporation, with an operating frequency range of 0.1 THz to 4 THz, and a maximum dynamic range of over 60 dB. The fabricated device was placed on an external heater with a hole in the center through which THz waves can be transmitted without loss. The THz transmitted wave was measured while raising the temperature of the heater on which the device was mounted to measure

the band-switching and tunable characteristics according to the change in the conductivity of the VO$_2$ thin film. The temperature of the heater was controlled by the applied voltage, and it directly affected the temperature of the VO$_2$ thin film; thus, the phase of the VO$_2$ thin film could be continuously changed from insulator to conductor.

Figure 7a,b show the THz time domain waveforms of the two metamaterials with different aspect ratios depending on the voltage applied to the heater in modes 1 and 2, respectively. The solid and dotted lines represent the measurement results of the metamaterials with the rectangular aspect ratios of 1.0 and 0.9, respectively. The THz wave transmittance was higher when the polarization direction of the incident THz wave was perpendicular to the etched VO$_2$ thin film than when it was horizontal. This is the same principle as that of a linear THz polarizer with a repetitive metal wire-grid structure that allows THz waves of polarization perpendicular to the line to pass through better than those of horizontal polarization [29]. As the voltage applied to the metamaterial increases and the conductivity of the VO$_2$ thin film increases, the transmittance of the THz wave gradually decreases. In addition, as the aspect ratio of the rectangle decreases, the ratio of the area of the metal structure to the total area of the metamaterial increases, such that the amount of THz wave transmission decreases.

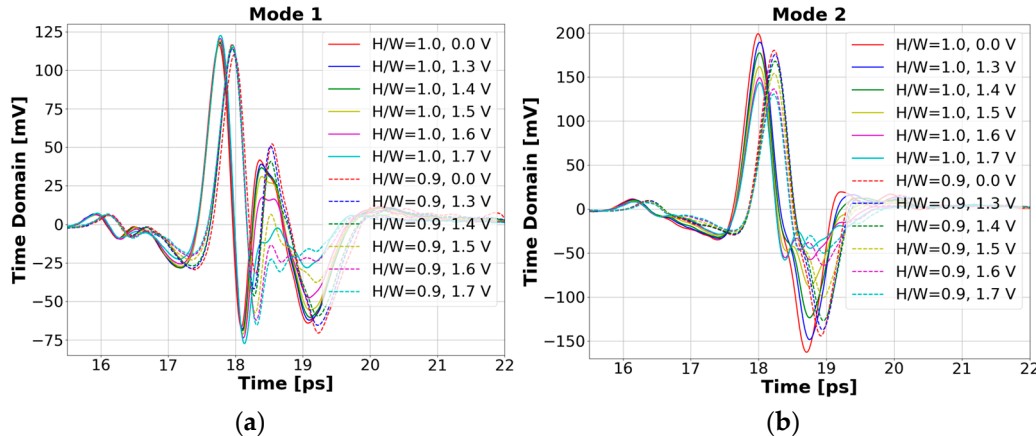

**Figure 7.** THz time-domain waveforms of two metamaterials with different aspect ratios depending on the voltage applied to the heater in (**a**) mode 1 and (**b**) mode 2.

Figure 8a,b show the THz transmittances of the two metamaterials with different aspect ratios, depending on the voltage applied to the heater in modes 1 and 2, respectively. As the applied voltage increases, the resonance characteristics of the metamaterials change in modes 1 and 2, from insulator state resonance to conductor state resonance. The resonance of the metamaterial is determined by the metal structure corresponding to the polarization of the incident THz wave. The horizontal and vertical metals of the double-split rectangle are the dominant resonant structures for mode 1 and mode 2, respectively. When the VO$_2$ thin film is converted into a conductor, there is almost no change in the resonant frequency, as shown in Figure 8a, because there is no change in electrical length in the case of mode 1. However, in mode 2, the resonant frequency is halved because the two vertical metals are electrically connected by the VO$_2$ thin film, as shown in Figure 8b. As shown in Figure 8a, in mode 1, as the applied voltage increases, the intensity of the primary resonance weakens, and the transmittance of the low- and high-frequency bands gradually decreases, suggesting that the central resonance shifts to the low- and high-frequency bands. However, in the experiment, the maximum conductivity of the VO$_2$ thin film was not sufficiently secured; hence, the simulation result could not confirm an evident resonance change, as shown in Figure 2. In Figure 8b, as the applied voltage increases in mode2, the switching characteristics of the 1.2 THz resonant frequency band transition to the 0.56 THz resonant frequency band were observed. Even in this case, the conductivity of the VO$_2$ thin film in the conductor state was insufficient; achieving a good quality factor was challenging as in

the simulation. In both modes, when the aspect ratio of the metamaterial decreases, the resonance band shifts to a higher frequency. The measurement results of the fabricated metamaterial obtained using the THz-TDS system showed a similar trend to the simulation results, except for the difference in the conductivity value of the $VO_2$ thin film.

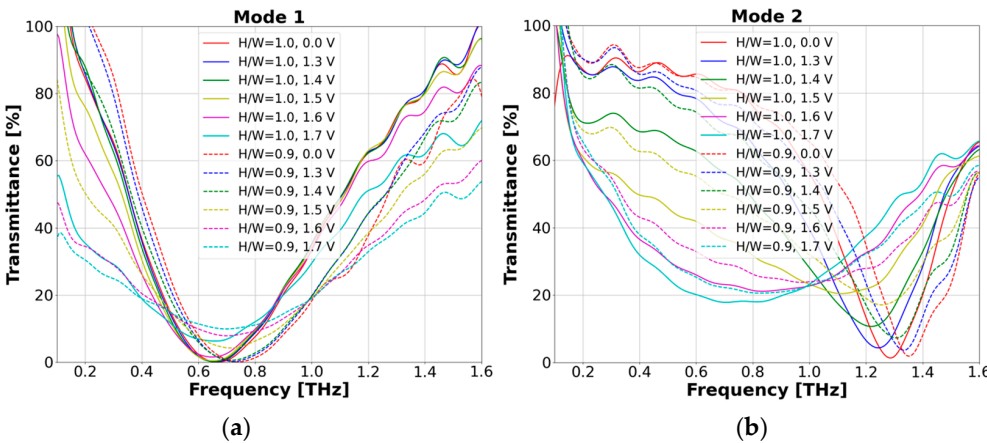

**Figure 8.** THz transmittance of the two metamaterials with different aspect ratios depending on the voltage applied to the heater in (**a**) mode 1 and (**b**) mode 2.

Figure 9a,b show the THz transmittance of the two metamaterials with different aspect ratios depending on the polarization of the incident THz wave when $VO_2$ is in the insulator and conductor states, respectively. When $VO_2$ is in the insulator state, lowering the aspect ratio of the rectangle reduces the physical length of the resonator because the width is fixed, thereby increasing the resonant frequency in both modes 1 and 2. In mode 1, resonance occurs in a resonator not separated by a gap, and in mode 2 it occurs in two resonators separated by gaps. Therefore, mode 1 resonance occurs in the 0.7 THz band, which is approximately half that in mode 2, where resonance occurs in the 1.3 THz band. As shown in Figure 9b, when $VO_2$ is in a conductor state, the resonance strength is weaker than the simulation result, resulting in a smaller quality factor. In particular, when the metamaterial operates in mode 1, the low-frequency resonance that appears strong in the simulation is not visible in the measurement results. This is because the low-frequency resonance generated in the $VO_2$ thin film line is weakened owing to insufficient conductivity of the fabricated $VO_2$ thin film. However, we can confirm the change in the resonance characteristics of the double-split rectangular metamaterial according to the increase in the conductivity of the $VO_2$ thin film, and the frequency blue-shift characteristic according to the decrease in the aspect ratio of the rectangle.

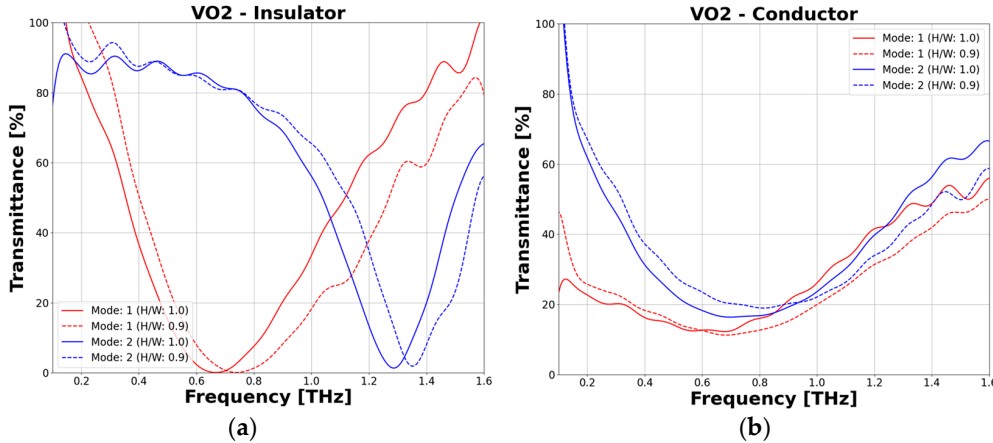

**Figure 9.** THz transmittance of the two metamaterials with different aspect ratios depending on the polarization of the incident THz wave when $VO_2$ is in the (**a**) insulator state and (**b**) conductor state.

## 4. Discussion

It was confirmed through simulation and measurement results that the resonance characteristics of a $VO_2$ thin film-based double-split rectangular metamaterial could be precisely controlled in the terahertz (THz) frequency domain. Since the direction of the THz electric field incident in mode 1 is the same as the direction of the etched $VO_2$ thin film line pattern, resonance mainly occurs due to the double-split rectangular metal structure parallel to the etched $VO_2$ thin film line. The electrical length of this horizontal metal structure does not change significantly even when the $VO_2$ thin film becomes a conductor. Therefore, even if the conductivity of the $VO_2$ thin film changes according to the change in applied voltage, there is no significant change in the resonance frequency. However, in mode 2, since the direction of the incident THz electric field is perpendicular to the line pattern direction of the etched $VO_2$ thin film, resonance mainly occurs due to the double-split rectangular vertical metal structure perpendicular to the etched $VO_2$ thin film. When the $VO_2$ film becomes a conductor, the two vertical metal structures are electrically connected, doubling their electrical length. Therefore, when the $VO_2$ thin film changes from an insulator to a conductor, the resonant frequency of the metamaterial is halved. Therefore, as the applied voltage increases, the resonant frequency shifts by half significantly.

The resonance characteristics were also successfully controlled by changing the aspect ratio and conductivity of the $VO_2$ thin film of the double-split rectangular metamaterial. As the aspect ratio decreased, the resonant frequency of the metamaterial increased, and the resonant band shifted gradually as the conductivity increased. As the conductivity was lower when the $VO_2$ thin film changed to a conductive state than that used in the simulation, the measured resonant strength of the metamaterial was lower than the simulation result. However, with the change in the conductivity of the $VO_2$ thin film, the resonance characteristics of the metamaterial changed in the same manner as the operating characteristics analyzed in the simulation. Our measurement results showed that the resonant frequency and quality factor of the double-split rectangular metamaterial based on the etched $VO_2$ thin film could be fine-tuned through the control of the aspect ratio. In addition, band switching and a gradual change in the resonance strength were possible through a change in the conductivity of the $VO_2$ thin film. The development of THz devices with fine-tunable the resonance characteristics is expected to be applicable to various applications, such as large-capacity THz tags, variable filters, multi-function sensors, and core parts of THz wireless communication.

## 5. Conclusions

We proposed a double-split rectangular metamaterial based on an etched $VO_2$ thin film with tunable resonance properties in the THz frequency band. We successfully fine-tuned the resonant frequency and strength of the metamaterials by controlling the aspect ratio of the rectangle. The loss of metamaterials was reduced using the etched $VO_2$ thin film, and gradual resonance-tuning characteristics were obtained by changing the conductivity of the thin film. The resonance control characteristics of the metamaterials obtained through the simulation were consistent with the measurement results obtained using the THz-TDS system. However, the conductivity of the $VO_2$ thin film was insufficient; therefore, the quality factor when the $VO_2$ thin film was in the conductive state was smaller than the simulation result. The quality factor of the metamaterials in which the $VO_2$ thin film is in the conductive state can be enhanced by improving the film quality or changing the unit cell structure. The resonance characteristics of the proposed THz device can finely and gradually tuned, increasing the capacity of THz tags, the usefulness of THz sensors, and the practical applicability of THz wireless communication.

**Author Contributions:** Conceptualization, E.S.L. and H.-C.R.; methodology, E.S.L. and H.-C.R.; software, H.-C.R.; writing—original draft preparation, E.S.L. and H.-C.R.; writing—review and editing, H.-C.R.; and supervision, H.-C.R. All authors have read and agreed to the published version of the manuscript.

**Funding:** This work was supported by Basic Science Research Program through the National Research Foundation (NRF) funded by the Ministry of Science and ICT (grant no. NRF-2021R1F1A1059493) and an Electronics and Telecommunications Research Institute (ETRI) grant funded by the Korea government (grant no. 22ZB1140, Development of Creative Technology for ICT).

**Institutional Review Board Statement:** Not applicable.

**Informed Consent Statement:** Not applicable.

**Data Availability Statement:** Data are contained within the article.

**Conflicts of Interest:** The authors declare that they have no known competing financial interests or personal relationships that could have influenced the research reported in this pape.

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
