# Peer review of "Resonance Control of VO2 Thin-Film-Based THz Double-Split Rectangular Metamaterial According to Aspect Ratio"

_photonics, doi:10.3390/photonics9120966_

Round 1

Reviewer 1 Report

In this manuscript, the authors proposed a double-split rectangular metamaterial based on the etched VO2 thin film which can control the resonant frequency and quality factor by changing the rectangle's aspect ratio. The results are of sufficient interest for researchers interested in the field of THz sensors. However, there are some confusing points in the manuscript, which need to be clarified. My comments are listed as follows:

1. The most important thing needing to be explained is that Fig. 3 and 5 are simulation results, while Fig. 9 is the experiment results. Although Fig. 3a, 5a are similar to Fig. 9a, Fig. 3b for mode 1 with conductive VO2 is very different with that in Fig. 9b. What is the reason?

2. In Fig. 3, the spectrum for VO2 insulator shows a dip, why does it become a peak for VO2 conductor? What is the physical mechanism?

3. Why there are black regions at the connections of VO2 and metal antennas in Fig. 2(d)?

4. In the part of "Measurement of a double-split rectangular metamaterial", in Mode 1 of FIG. 8(a), the response frequency shifts very little as the voltage increases. However, for mode 2 in Fig.8(b), with the increase of voltage, the response frequency moves from ~1.3THz to 0.7THz, which is very close to the simulation result. Therefore, the conductivity of VO2 should be sufficient to achieve resonant frequency modulation. Thus, why is the modulation in mode 1 so small.

5. The permittivity or dielectric constant of VO2 dependent on temperature is not given.

Author Response

----------------------------------------------------------------------------------------------------------------Response to Reviewers: First of all, we appreciate your comments and considerations for our manuscript. We found that the comments are useful in improving the clarity of our paper. We copy the reviewers’ comments here and use the italic font to differentiate them from our reply. Our reply to each comment of the reviewers is as below.                                                        

----------------------------------------------------------------------------------------------------------------

-------- Reviewer Comments --------

Reviewer 1:
In this manuscript, the authors proposed a double-split rectangular metamaterial based on the etched VO2 thin film which can control the resonant frequency and quality factor by changing the rectangle's aspect ratio. The results are of sufficient interest for researchers interested in the field of THz sensors. However, there are some confusing points in the manuscript, which need to be clarified. My comments are listed as follows:

 Question 1. The most important thing needing to be explained is that Fig. 3 and 5 are simulation results, while Fig. 9 is the experiment results. Although Fig. 3a, 5a are similar to Fig. 9a, Fig. 3b for mode 1 with conductive VO2 is very different with that in Fig. 9b. What is the reason?

[Reply 1] We sincerely thank the reviewer for providing valuable and helpful comments on the reasons for the discrepancy between simulation and experimental results. First, I would like to note that the figure number has been modified based on comments from another reviewer. Figures 3 and 5 have been changed to Figures 2 and 4, and Figure 9 is unchanged. What reviewers noted was what we struggled with the most. When the VO2 thin film is in a conductor state, the simulation shows a strong low-frequency resonance responding to the VO2 line, which is a long conductor, as shown in Figure 2(b), but the measurement does not show a strong low-frequency resonance. This is understood to be a phenomenon that occurs because the conductivity of the VO2 thin film in the conductor state is not sufficient. The simulation result of the transmission characteristics changes according to the conductivity change of the VO2 thin film was presented in our previous paper (Ref. 21) (see the figure below). It can be confirmed that the simulation results when the conductivity of the VO2 thin film is 5e4 ~ 1e5 S/m are similar to the measurement results of this paper. The simulation result when VO2 is a conductor, shown in Figure 2(b), is the simulation result when the conductivity of the VO2 thin film is 1e6 S/m.

(Ref. 21 - Shin, J. H.; Park, K. H.; Ryu, H. C. A band-switchable and tunable THz metamaterial based on an etched vanadium dioxide thin film. Photonics 2022, 9, 89.)

In this regard, we added the following sentences:

The conductivity of the VO2 thin film was set to 20 S/m in the insulator state and 1,000,000 S/m in the conductor state.”  (Page 3, line 121-123 in Design and Simulation)

As shown in Fig. 9(b), when VO2 is in a conductor state, the resonance strength is weaker than the simulation result, resulting in a smaller quality factor. In particular, when the metamaterial operates in mode 1, the low-frequency resonance that appears strong in the simulation is not visible in the measurement results. This is because low-frequency resonance generated in the VO2 thin film line is weakened owing to insufficient conductivity of the fabricated VO2 thin film.”  (Page 10, line 346-351 in Fabrication and Measurement)

Question 2. In Fig. 3, the spectrum for VO2 insulator shows a dip, why does it become a peak for VO2 conductor? What is the physical mechanism?

[Reply 2] We thank the reviewer for valuable comments. Figure 3 has been changed to Figure 2 based on comments from another reviewer. There are two reasons why the dip in the 0.55-0.85 THz band changed to the peak in the 1.0-1.4 THz band. The first is due to the low-frequency resonance caused by the VO2 thin line in the conductor state. The second is that the 0.55 to 0.85 THz band resonance shifts to the 1.25 to 1.6 THz band resonance due to the conductive properties of the VO thin film. For the analysis of resonance band shift, surface current density was compared and analyzed in Figure 3. And we added the following sentence:

“Moreover, a new resonance occurred in the low-frequency band, changing the dip in the 0.55 to 0.85 THz band to the peak in the 1.0 to 1.4 THz band.” (Page 4, lines 170 - 172 in Design and Simulation)

Question 3. Why there are black regions at the connections of VO2 and metal antennas in Fig. 2(d)?

[Reply 3] We thank the reviewer for your valuable comments.  Figure 2 has been changed to Figure 6 based on comments from another reviewer. The overlapping parts between the VO2 thin film line and the gold pattern appear black. Figures 6(b) and 6(d) look different because the microscope focus and light intensity are different when measuring each device. Although many efforts were made to obtain clear images under the same conditions, it was impossible to obtain precisely the same images at 80x magnification.

Question 4. In the part of "Measurement of a double-split rectangular metamaterial", in Mode 1 of FIG. 8(a), the response frequency shifts very little as the voltage increases. However, for mode 2 in Fig.8(b), with the increase of voltage, the response frequency moves from ~1.3THz to 0.7THz, which is very close to the simulation result. Therefore, the conductivity of VO2 should be sufficient to achieve resonant frequency modulation. Thus, why is the modulation in mode 1 so small.

[Reply 4] We thank the reviewer for the valuable comment on the tunability of the metamaterial. This question is very closely related to the first question. Since the direction of the incident THz electric field in mode 1 is the same as the direction of the etched VO2 thin film line pattern, resonance mainly occurs due to the double-split rectangular horizontal metal structure parallel to the etched VO2 thin film line. The electrical length of this horizontal metal structure does not change significantly even when the VO2 thin film becomes a conductor. Therefore, even if the conductivity of the VO2 thin film changes according to the change in applied voltage, there is no significant change in the resonant frequency. However, in mode 2, since the direction of the incident THz electric field is perpendicular to the line pattern direction of the etched VO2 thin film, resonance mainly occurs due to the double-split rectangular vertical metal structure perpendicular to the etched VO2 thin film. When the VO2 thin film changes to a conductor, the two vertical metal structures are electrically connected, doubling their electrical length. Therefore, when the VO2 thin film changes from an insulator to a conductor, the resonant frequency of the metamaterial is halved. Therefore, as the applied voltage increases, the resonant frequency significantly shifts by half.

“The resonance of the metamaterial is determined by the metal structure corresponding to the polarization of the incident THz wave. The horizontal and vertical metals of the double-split rectangle are the dominant resonant structures for mode 1 and mode 2, respectively. When the VO2 thin film is converted into a conductor, there is almost no change in the resonant frequency, as shown in Figure 8(a), because there is no change in electrical length in the case of mode 1. However, in mode 2, the resonant frequency is halved because the two vertical metals are electrically connected by the VO2 thin film, as shown in Figure 8(b).”  (Page 9, lines 312 - 319 in Fabrication and Measurement)

Question 5. The permittivity or dielectric constant of VO2 dependent on temperature is not given.

[Reply 5] We thank the reviewers for their valuable comments on the temperature dependence of the VO2 dielectric constant. A previous paper reported the change in conductivity of VO thin films with temperature. In the simulation, the change in the transmission properties of the metamaterial was calculated while the permittivity of VO2 was fixed and the conductivity was changed. I've added and referenced the following sentence for clarity.

“The conductivity of the VO2 thin film was set to 20 S/m in the insulator state and 1,000,000 S/m in the conductor state. The conductivity was confirmed by measurement using a four-probe measurement [24]. The dielectric constants of the VO2 thin film and Al2O3 were fixed at 9.1 and 9.4, respectively.” (Page 3, lines 121 - 125 in Design and Simulation)

(Ref. 24 - Shin, J. H.; Park, K. H.; Ryu, H. C. Electrically controllable terahertz square-loop metamaterial based on VO2 thin film. Nanotechnology, 2016, 27, 195202)

Reviewer 2 Report

In this thesis, the authors propose a double split rectangular metamaterial based on etching VO2 film, which has controllable resonance characteristics in terahertz band. By controlling the aspect ratio of the rectangle, the authors successfully fine tuned the resonant frequency and intensity of the metamaterial. A double-split rectangular metamaterial proposed in this study could stably control the resonant frequency and resonance strength of the device only through the aspect ratio change, not the overall unit-cell size change. The proposed terahertz device can fine-tune and gradually tune the resonance characteristics, increasing THz tags' capacity, the usefulness of THz sensors, and the practical applicability of THz wireless communication. I believe that publication of the manuscript may be considered only after the following issues have been resolved.

1.       The author needs to make major adjustments to theDesign and Fabrication” part of the article. First is the simulation part. The author needs to give more detailed parameters, including calculation methods, material parameters, etc. Then comes the preparation part. The author needs to put Figure 2 in the results and discussion part. In addition, the author needs to give the specific preparation process and method in the preparation part, so as to give readers space for repetition.

2.       The text information in Figure 2 is not clear.

3.       How to realize the simulation of the two vanadium oxide states, insulator and conductor, involved in Figure 6? The author needs to provide specific simulation parameters.

4.       The introduction can be improved. The articles related to some applications of vanadium dioxide materials should be added such as Phys. Chem. Chem. Phys., 2022, 24, 8846 – 8853; Physical Chemistry Chemical Physics, 2022, 24, 2527 – 2533.

5.       About “simulation part”, some articles related to physical models need to be mentioned, such as Plasmonics 2015, 10, 1537–1543; Plasmonics 2018, 13, 345–352.

6.       Please check the grammar and spelling mistakes of the whole manuscript.

Author Response

----------------------------------------------------------------------------------------------------------------Response to Reviewers: First of all, we appreciate your comments and considerations for our manuscript. We found that the comments are useful in improving the clarity of our paper. We copy the reviewers’ comments here and use the italic font to differentiate them from our reply. Our reply to each comment of the reviewers is as below.                                                        

----------------------------------------------------------------------------------------------------------------

-------- Reviewer Comments --------

Reviewer 2:

In this thesis, the authors propose a double split rectangular metamaterial based on etching VO2 film, which has controllable resonance characteristics in terahertz band. By controlling the aspect ratio of the rectangle, the authors successfully fine tuned the resonant frequency and intensity of the metamaterial. A double-split rectangular metamaterial proposed in this study could stably control the resonant frequency and resonance strength of the device only through the aspect ratio change, not the overall unit-cell size change. The proposed terahertz device can fine-tune and gradually tune the resonance characteristics, increasing THz tags' capacity, the usefulness of THz sensors, and the practical applicability of THz wireless communication. I believe that publication of the manuscript may be considered only after the following issues have been resolved.

Question 1. The author needs to make major adjustments to the “Design and Fabrication” part of the article. First is the simulation part. The author needs to give more detailed parameters, including calculation methods, material parameters, etc. Then comes the preparation part. The author needs to put Figure 2 in the results and discussion part. In addition, the author needs to give the specific preparation process and method in the preparation part, so as to give readers space for repetition.

[Reply 1] We appreciate the reviewer for valuable comments. We changed the "Fabrication" part to behind the "simulation". And, the material parameters and calculation methods were additionally described in the beginning part of the simulation.

“The THz transmission characteristics of the proposed metamaterial depend on the change in the conductivity of the VO2 thin film, and the conductivity of the VO2 thin film depends on the temperature. The conductivity of the VO2 thin film was set to 20 S/m in the insulator state and 1,000,000 S/m in the conductor state. The conductivity was confirmed by measurement using a four-probe measurement [24]. The dielectric constants of the VO2 thin film and Al2O3 were fixed at 9.1 and 9.4, respectively. The frequency dependence of the material is not considered.” (Page 3, lines 119-125 in Design and Simulation)

“The port was set so that only one mode was incident with a horizontal (parallel to the etched VO2 line) and vertical (perpendicular to the etched VO2 line) polarization electric field for mode 1 and mode 2, respectively.” (Page 3, lines 133-135 in Design and Simulation)

Question 2. The text information in Figure 2 is not clear.

[Reply 2] We would like to thank the reviewer for his or her helpful comments on the clarity of our paper. First, I would like to note that Figure 2 has been changed to Figure 6 based on previous comments from reviewers. And, we added some text information related to Figure 6.

“Figure 6(a) and 6(b) are photographs of a metamaterial with a rectangle aspect ratio of 1.0 at 20x and 80x magnification, respectively, and Figure 6(c) and 6(d) are photographs with an aspect ratio of 0.9 at 20x and 80x magnification, respectively.” (Page 7, lines 254-257 in Fabrication and Measurement)

Question 3. How to realize the simulation of the two vanadium oxide states, insulator and conductor, involved in Figure 6? The author needs to provide specific simulation parameters.

 [Reply 3] We appreciate the reviewer for valuable comments. Figure 6 was changed to Figure 5 based on comments from the reviewer. We added the simulation parameter for the two vanadium dioxide states, insulator and conductor, as below:

The conductivity of the VO2 thin film was set to 20 S/m in the insulator state and 1,000,000 S/m in the conductor state. The conductivity was confirmed by measurement using a four-probe measurement [24]. The dielectric constants of the VO2 thin film and Al2O3 were fixed at 9.1 and 9.4, respectively. The frequency dependence of the material is not considered.” (Page 3, lines 121-125 in Design and Simulation)

Question 4. The introduction can be improved. The articles related to some applications of vanadium dioxide materials should be added such as Phys. Chem. Chem. Phys., 2022, 24, 8846 – 8853; Physical Chemistry Chemical Physics, 2022, 24, 2527 – 2533.

[Reply 4] We thank the reviewer for valuable comments. we added two references recommended by the reviewer like below.

  1. Zheng, Z.; Zheng, Y.; Luo, Y.; Yi, Z.; Zhang, J.; Liu, Z.; Yang, W.; Yu, Y.; Wu, X.; Wu, P. A switchable terahertz device combining ultra-wideband absorption and ultra-wideband complete reflection. Phys. Chem. Chem. Phys., 2022, 24, 2527-2533.

  1. Zheng, Z; Luo, Y.; Yang, H.; Yi, Z.; Zhang, J.; Song Q.; Yang, W.; Liu, C.; Wu, X.; Wu, P. Thermal tuning of terahertz metamaterial absorber properties based on VO2. Phys. Chem. Chem. Phys., 2022, 24, 8846-8853.

Question 5. About “simulation part”, some articles related to physical models need to be mentioned, such as Plasmonics 2015, 10, 1537–1543; Plasmonics 2018, 13, 345–352.

[Reply 5] We thank the reviewer for valuable comments. The content and topic of the paper recommended by the reviewer are significant, but it is challenging to mention the content because it is a little far from this paper. No coupling or plasmonic theory was applied when interpreting the metamaterial presented in this paper.

Question 6. Please check the grammar and spelling mistakes of the whole manuscript.

[Reply 6] We thank the reviewers for his or her helpful comments on the clarity of the paper. In order to correct the grammar and spelling of the whole manuscript, we received a proofreading service from a native speaker and proofread the whole manuscript.

Round 2

Reviewer 1 Report

The authors have responded all my questions and made corresponding improvements. Thus I recommend the manuscript can be considered to be accepted by Photonics.

Reviewer 2 Report

Accept in present form.